# A novel and effective method for solving the router nodes placement in wireless mesh networks using reinforcement learning

Le Huu Binh[1], Thuy-Van T. Duong[2]*

1 Faculty of Information Technology, University of Sciences, Hue University, Hue City, Vietnam, 2 Faculty of Information Technology, Ton Duc Thang University, Ho Chi Minh City, Viet Nam

◉ These authors contributed equally to this work.
* duongthithuyvan@tdtu.edu.vn

**Data Availability Statement:** All relevant data are within the manuscript and its Supporting information files.

## Abstract

Router nodes placement (RNP) is an important issue in the design and implementation of wireless mesh networks (WMN). This is known as an P-hard problem, which cannot be solved using conventional algorithms. Consequently, approximate optimization strategies are commonly used to solve this problem. With heavy node density and wide-area WMNs, solving the RNP problem using approximation algorithms often faces many difficulties, therefore, a more effective solution is necessary. This motivated us to conduct this work. We propose a new method for solving the RNP problem using reinforcement learning (RL). The RNP problem is modeled as an RL model with environment, agent, action, and reward are equivalent to the network system, routers, coordinate adjustment, and connectivity of the RNP problem, respectively. To the best of our knowledge, this is the first study that applies RL to solve the RNP problem. The experimental results showed that the proposed method increased the network connectivity by up to 22.73% compared to the most recent methods.

## Introduction

Wireless communication is growing and being widely applied in many fields. In the local area network of agencies, businesses, schools, and so on, wireless mesh networks (WMN) [1, 2] are the best choice today because of their significant advantages compared to wireless networks using traditional access points. The most notable benefit of the WMN is that it reduces congestion owing to its ability to balance the loads. In addition, the installation of a WMN is very convenient because there is no need to construct wired connections from the gateway to all routers. Fig 1 illustrates an example of a WMN consisting of six mesh routers (represented by $r_1$ to $r_6$) and eleven mesh clients (represented by $c_1$ to $c_{11}$). In addition, at least one the router of the Internet service provider serves as a gateway for clients to access the Internet. If two mesh routers are within range of each other, a wireless link is established between them. A mesh topology consists of of all the mesh routers and wireless links. For a WMN to deliver Internet services, several mesh routers must be connected to the gateway router via wireless or

**Funding:** The authors received no specific funding for this work.

**Competing interests:** The authors have declared that no competing interests exist.

cable links. As shown in Fig 1, the mesh routers $r_1$ and $r_2$ are connected to the gateway router (GPON or FTTh router) via wireless links. Mesh clients are terminal devices that are users of network services. When a mesh client enters the network region, it can be covered by one or more mesh routers; the mesh client connects to the nearest mesh router to access network services.

With the rapid development of wireless and mobile communication technologies, network services are becoming more diverse and rich, especially those on fifth-generation (5G) and sixth-generation (6G) wireless network platforms. To effectively provide these services, WMNs must be designed and installed in the most efficient manner possible, allowing network resources to be fully utilized. This is the motivation for researchers to focus on WMN. Some of the most prevalent subjects that have been implemented include network topology control [3–7], router node placement (RNP) [8–24], optimum routing protocols [25–29], and access point allocation [30–33], with the RNP challenge being the most fascinating. Because the RNP problem is known to be NP-hard, it cannot be solved using conventional algorithms. Recently, approximate optimization methods have become useful for solving this problem [8–12]. The authors of [8] have used the coyote optimization algorithm (COA) to solve the RNP problem. Their proposed method optimizes both network connectivity and user coverage, which are two critical performance criteria. Using MATLAB simulations, the authors demonstrated that the COA algorithm outperformed other well-known optimization algorithms. In [10], the authors suggested an optimal method called the Chemical Reaction Optimization (CRO) algorithm to solve this problem. The CRO algorithm was inspired by how molecules interact to achieve a low, stable energy state in chemical reactions. In terms of client coverage and network connection, the simulation findings reveal that their suggested approach outperforms the Genetic approach (GA) and Simulated Annealing (SA). Another study employed a genetic algorithm and simulated annealing to discover a low-cost WMN configuration while satisfying restrictions and identifying the number of gateways needed [34]. Experiments showed that the evolutionary algorithm and simulated annealing were successful in lowering WMN network

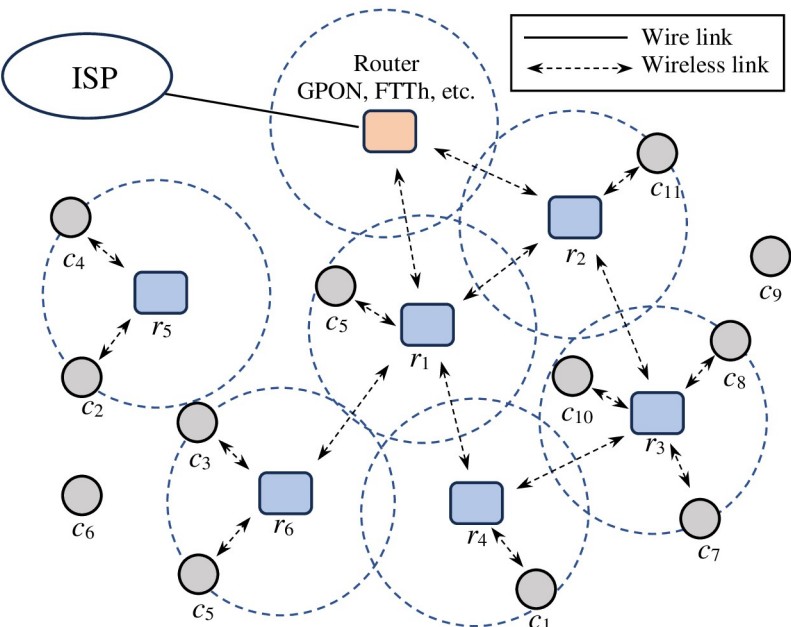

**Fig 1. An example of a wireless mesh network.**

expenses while maintaining QoS. The new models significantly outperformed the conventional solutions. QoS was also considered in the RNP problem in [23]. The authors described a unique particle swarm optimization method for improving network connectivity and client coverage. The QoS restrictions for this study are the delay, relay load, and Internet gateway capacity. In [35], the authors suggested an improved version of the Moth Flame Optimization (MFO) algorithm, namely, Enhanced Chaotic Lévy Opposition-based MFO (ECLO-MFO), for solving the RNP problem. To improve the optimization performance of MFO, the proposed method integrates three strategies: the chaotic map concept, Lévy flying strategy, and Opposition-Based Learning (OBL) technique. The simulation results showed that the proposed algorithm was more efficient than the method of applying popular optimization algorithms.

Based on the results of published works, we find that the method of using approximate optimal algorithms provide good solutions. However, because randomness is used in several steps of the algorithm, the results often differ for different executions. For accurate results, each script must be executed multiple times, and then the average of all executions is obtained. For example, the authors of [8, 11] executed each simulation scenario 50 times. Furthermore, with heavy node density and wide-area WMNs, solving the RNP problem with approximation algorithms often presents many difficulties, necessitating a more effective solution. In this paper, we propose a new and effective algorithm to solve this problem. The main contributions of this study are summarized as follows:

(i) We proposed a novel and effective method for solving the RNP problem using RL. The RNP problem is modeled as an RL model, with the environment, agent, action, and reward representing the network system, routers, coordinate adjustment, and connectivity respectively, of the RNP problem. To the best of our knowledge, this is the first study to apply reinforcement learning to the RNP problem.

(ii) We compared and evaluated the performance of the RNP problem solving method using the heuristic algorithms and the RL method.

The remainder of this paper is organized as follows. The next section describes the formulation of the RNP problem in the WMN. The following sections present our proposed solution and experimental results. Finally, concluding remarks and promising future studies are presented in the last section.

## RNP problem

In this section, we formulate the RNP problem in a WMN. First, graph theory was used to describe the WMN. We then define some metrics to use for the objective function of the RNP problem, similar to [11]. Finally, the RNP problem was formulated as a nonlinear programming problem. For convenience, we define the mathematical symbols shown in Table 1.

### Mathematical model of a WMN using graph theory

Consider a WMN comprising $m$ mesh routers, $n$ mesh clients, and $k$ gateway routers. Mathematically, this WMN can be represented as an undirected graph, denoted by $G = (V, E)$, where $V$ and $E$ are the vertex and edge sets, respectively. $V$ is equivalent to the set of all nodes in the WMN and is determined by $V = R \cup C \cup W$, where $R$, $C$ and $W$ are the sets of mesh routers, mesh clients, and gateway routers, respectively. $E$ is equivalent to the set of all wireless links in the WMN and consists of three types: links between mesh routers, links between mesh client and mesh router, and links between gateway and mesh router.

**Table 1. The notations used in this paper.**

| Notation | Description |
|---|---|
| $m$ | Number of mesh routers |
| $n$ | Number of mesh clients |
| $k$ | Number of gateway routers |
| $R = \{r_i \| i = 1..m\}$ | Set of mesh routers |
| $C = \{c_i \| i = 1..n\}$ | Set of mesh clients |
| $G = \{g_i \| i = 1..k\}$ | Set of gateway routers |
| $P_r = \{(x_{r_i}, y_{r_i}) \| i = 1..m\}$ | Set of the coordinates of mesh routers |
| $P_c = \{(x_{c_i}, c_{r_i}) \| i = 1..n\}$ | Set of the coordinates of mesh clients |
| $P_g = \{(x_{g_i}, y_{g_i}) \| i = 1..k\}$ | Set of the coordinates of gateway routers |
| $T = (V, E)$ | Graph representing the topology of WMN |
| $V = R \cup C \cup G$ | Set of nodes in WMN |
| $E$ | Set of wireless links in WMN |
| $W$ | The width of the WMN area |
| $H$ | The height of the WMN area |
| $d(c_i, r_j)$ | Distance between client $c_i$ and router $r_j$ |
| $d(r_i, r_j)$ | Distance between routers $r_i$ and $r_j$ |
| $\lambda$ | Parameters control the metrics |
| $d_r$ | Coverage radius of mesh routers |
| $\alpha(r_i)$ | A function that returns 1 if the mesh router $r_i$ is a connected router, otherwise 0 |
| $\beta(c_i)$ | A function that returns 1 if the mesh client $c_i$ is a connected client, otherwise 0 |
| $CRR$ | Connected router ratio |
| $CCR$ | Connected client ratio |
| $s_t$ | The state of the environment at time $t$ |
| $A_t$ | Set of actions at time $t$ |
| $RW(r_i, s_t, a_{t,k})$ | Reward obtained when mesh router $r_i$ performs the action $a_{t,k} \in A_t$ at the state $s_t$ |
| $Q(r_i, s_t, a_{t,k})$ | Q-Value when mesh router $r_i$ performs the action $a_{t,k} \in A_t$ at the state $s_t$ |
| $\pi(r_i, s_t, a_{t,k})$ | Probability of mesh router $r_i$ chooses action $a_{t,k} \in A_t$ at the state $s_t$ using $\varepsilon\text{-}greedy$ policy |

## RNP problem formulation

In this section, we formulate the RNP problem using some concepts and metrics from [11], including the connected router, connected client, connected router ratio, and connected client ratio.

**Connected router.** The mesh router $r_i$ is a connected router if and only if at least one path exists between it and the gateway router. If we return to the WMN example in Fig 1, we can see that mesh routers $r_1$, $r_2$, $r_3$, $r_4$ and $r_6$ are the connected routers but $r_5$ is not because no path exists from this mesh router to the gateway router.

**Connected router ratio (CRR).** The CRR is defined as the percentage of connected routers in relation to the total number of routers in a WMN, calculated by [11]

$$CRR = \frac{\sum_{i=1}^{m} \alpha(r_i)}{m} \times 100(\%) \tag{1}$$

where $m$ is the number of routers in a WMN and $\alpha(r_i)$ is a function that indicates whether

router $r_i$ is a connected router or no, defined by

$$\alpha(r_i) = \begin{cases} 1 & \text{if } r_i \text{ is a connected router} \\ 0 & \text{otherwise} \end{cases} \tag{2}$$

**Connected client.**  Mesh client $c_i$ is a connected client if and only if it is covered by at least one connected router. Let $\beta(c_i)$ be a function that indicates whether client $c_i$ is a connected client, returning 1 if yes and 0 otherwise. Then, $\beta(c_i)$ is calculated as

$$\beta(c_i) = \begin{cases} 1 & \text{if } \min_{j=1..m} d(c_i, r_j) \leq d_r \\ 0 & \text{otherwise} \end{cases} \tag{3}$$

where $d_r$ is the coverage radius of the routers, $d(c_i, r_j)$ is the distance between client $c_i$ and router $r_j$, given by

$$d(c_i, r_j) = \sqrt{(x_{c_i} - x_{r_j})^2 + (y_{c_i} - y_{r_j})^2} \tag{4}$$

where $(x_{c_i}, y_{c_i})$ and $(x_{r_j}, y_{r_j})$ are the coordinates of the client $c_i$ and router $r_j$, respectively. Considering the example of WMN in Fig 1, we can easily observe that the set of connected clients are listed as $\{c_1, c_3, c_5, c_6, c_7, c_8, c_{10}, c_{11}\}$. Client $c_9$ is not a connected client because it is not covered by any mesh router. For clients $c_2$ and $c_4$, although they are covered by router $r_5$, they are not connected clients, because $r_5$ is not the connected router.

**Connected client ratio (CCR).**  The CCR is defined as the percentage of the connected clients in relation to the total number of clients in a WMN, calculated by [11]

$$CCR = \frac{\sum_{i=1}^{n} \beta(c_i)}{n} \times 100(\%) \tag{5}$$

where $n$ is the number of mesh clients and $\beta(c_i)$ is determined according to (3).

**Formulate the RNP into a nonlinear programming problem.**  The RNP problem in the WMN is stated as follows: Consider a case where it is necessary to design and install a WMN with the following assumptions:

- The network system is located in an area of $W \times H$ meters.

- The number of clients is $n$, and they are located at a given set of coordinates $P_c = \{(x_{c_i}, y_{c_i}) | i = 1..n\}$.

- The number of gateway routers is $k$, they are located at a given set of coordinates $P_w = \{(x_{w_i}, y_{w_i}) | i = 1..k\}$, and the coverage radius of each gateway router is $d_w$.

- The number of mesh routers was $m$, and the coverage radius of each mesh router was $d_r$.

Find the set of coordinates $P_r = \{(x_{r_i}, y_{r_i}) | i = 1..m\}$ to place $m$ routers such that $CRR$ and $CCR$ are at their maximum. Thus, the NRP problem can be described as the following nonlinear programming problem:

$$\begin{cases} \textit{Maximize} \quad CRR \\ \textit{Maximize} \quad CCR \end{cases} \tag{6}$$

subject to the following constraints:

$$0 < x_{r_i} < W, \quad (i = 1..m) \tag{7}$$

$$0 < y_{r_i} < H, \quad (i = 1..m) \tag{8}$$

where $W$ and $H$ are the width and height of the network, respectively. By solving the nonlinear programming problem with objective functions (6) and the constraints (7) and (8), we find the coordinate set $P_r = \{(x_{r_i}, y_{r_i}) | i = 1..m\}$ to place $m$ mesh routers in the network area $W \times H$. This nonlinear programming problem can be solved in various ways. Recently, the method of applying approximate optimization algorithms to solve this problem has become popular [8–12]. In this work, we propose a new and effective method to solve this problem using reinforcement learning. The following sections describe this new method in detail.

## RL-based mesh router nodes placement

### Fundamentals of RL

A type of machine learning is called RL, in which the system learns from its past actions to choose wiser ones in the future. Fig 2 depicts the fundamental principles of RL, in which an agent operates as a learner, interacts with the environment to gain a reward and changes the state of the environment. At time $t$, the agent interacts with the environment through $a_t$ action. The environment changes from $s_t$ state to $s_{t+1}$ state as a result of this activity, and the agent is rewarded with an $r_t$. Based on the rewards acquired in the prior learning, the agent selects the action that provides the best reward in the following learnings. The total reward for taking the $a_t$ action in $s_t$ state is $Q(s_t, a_t)$, which is typically determined by the Q-learning algorithm as follows [36]:

$$Q(s_t, a_t) = (1 - \alpha)Q(s_t, a_t) + \alpha[R(s_t, a_t) + \gamma \underset{\forall a_{t+1} \in A}{Max} Q(s_{t+1}, a_{t+1})] \tag{9}$$

where $\alpha$ and $\gamma \in [0, 1]$ are the learning rate and the discount factors, respectively.

RL has been successfully applied to control protocols in wireless networks, typically routing in WMN [25, 27, 29], topology control in wireless sensor networks [37], improving the performance of energy-harvesting wireless body area networks [38, 39]. In this paper, we apply RL to solve the RNP in WMN. Details of this new proposal are presented in the following sections.

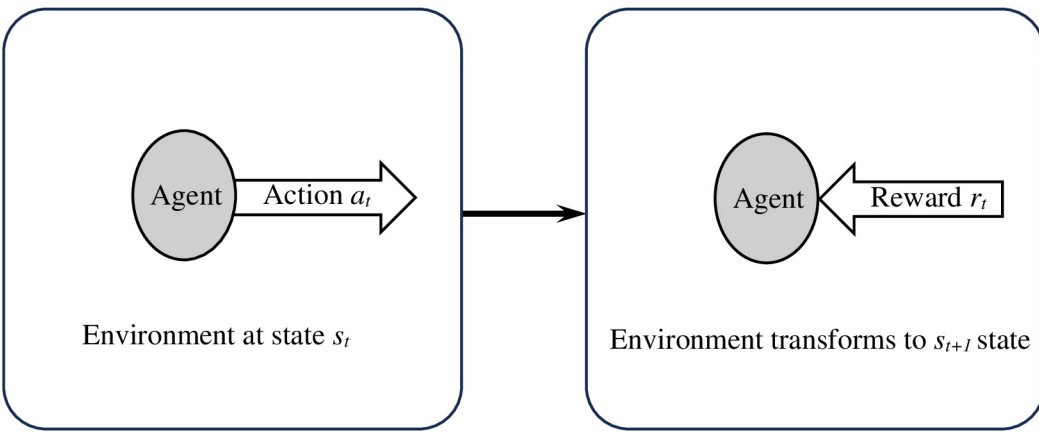

**Fig 2. Demonstrate the fundamental principles of RL.**

## Solving the RNP in WMN using RL

The RL has recently been successfully employed to solve technical challenges in wireless communication such as routing [27, 36], topology management [37], and resource allocation. In this study, we use RL to solve the RNP problem. To the best of our knowledge, this is the first study to use RL to address the RNP problem. To do this, the RNP problem must be modeled as a reinforcement learning model with five characteristic factors: agent, environment, state, action, and reward.

**Agent.**  An agent is a mesh router that regularly adjusts its coordinates to obtain an optimal topology.

**Environment.**  In a RL model, the environment is everything that exists around the agent, and it is where the agent acts and interacts. The environment for the RNP problem using RL is the network system, which includes a set of mesh routers, clients, gateway routers, and network area.

**State.**  Each state is determined by a triple $\{P_c, P_r, P_w\}$, where $P_c$, $P_r$ and $P_w$ are the sets of coordinates for the mesh clients, mesh router, and gateway routers, respectively. The sets are listed in Table 1.

**Action.**  Action is the way in which the agent interacts with the environment to change its state. For the RNP problem using reinforcement learning, the agents are the mesh routers. Each action was defined by a mesh router that adjusted its coordinates. The set of actions at a specific state $s_t$ for each mesh router $r_i$ is defined as $A_t$ = {*mn1s, me1s, ms1s, mw1s, mn2s, me2s, ms2s, mw2s*}, where the actions are described in Table 2, *step* is a given distance.

**Reward.**  The agent receives a reward for each action that interacting with the environment. The agent chooses the next action based on the reward value of past actions, with the goal of eventually achieving the best reward. For the RNP problem using reinforcement learning, we used the objective function defined in (6) as the reward for the learning process. This objective function consists of two metrics: *CRR* and *CCR*. To maximize both these metrics, we define the reward function as follows:

$$RW(r_i, s_t, a_t) = \lambda \times CRR_t + (1 - \lambda) \times CCR_t \qquad (10)$$

where $RW(r_i, s_t, a_t)$ is the reward obtained when the mesh router $r_i$ performs the action $a_t \in A_t$ at state $s_t$, $CRR_t$ and $CCR_t$ are the connected router ratio and the connected client ratio at state $s_t$, calculated according to (1) and (5), respectively. $\lambda$ is a coefficient in the range [0, 1], that is used to control the optimal degree of the metrics. In this study, the Q-learning algorithm is used to update the total reward each time a mesh router performs an action. Let $Q(r_i, s_t, a_t)$ be the total reward received after the mesh router $r_i$ performs the action $a_t \in A_t$ at state $s_t$, then $Q$

**Table 2. Actions taken by the mesh routers.**

| Notation | Description |
| --- | --- |
| *mn1s* | Move north one step |
| *me1s* | Move east one step |
| *ms1s* | Move south one step |
| *mw1s* | Move west one step |
| *mn2s* | Move north two steps |
| *me2s* | Move east two steps |
| *ms2s* | Move south two steps |
| *mw2s* | Move west two steps |

$(r_i, s_t, a_t)$ is given by

$$Q(r_i, s_t, a_t) = (1-\alpha)Q(r_i, s_t, a_t) + \alpha[RW(r_i, s_t, a_t) + \gamma \underset{\substack{\forall r_j \in R, \\ \forall a_{t+1} \in A_{t+1}}}{Max} Q(r_j, s_{t+1}, a_{t+1})] \tag{11}$$

where $\alpha$ and $\gamma \in [0, 1]$ are the learning rate and the discount factors, respectively.

**RL algorithm for solving RNP problem.** Algorithm 1 is the pseudo code of the RL algorithm for solving the RNP problem in the WMN. First, $m$ routers are placed at random coordinates in a network area of $W \times H$ [m] (step 1). For each learning time, the mesh router $r_i$ was randomly selected from set $R$ to perform an action in set $A_t$. The policy for selecting an action $a_k$ in set $A_t$ is $\varepsilon$ -greedy as in [37]. For this policy, the mesh router $r_i$ chooses action $a_t$ at state $s_t$ with a high probability of $1 - \varepsilon$ if the $Q(r_i, s_t, a_t)$ the value is maximum. The remaining actions in set $A_t$ are chosen with an equally low probability $\varepsilon$ (step 7), where $\varepsilon$ is set to 0.1, as in [37]. Let $\pi(r_i, s_t, a_{t,k})$ be the probability that the mesh router $r_i$ chooses action $a_{t,k}$ at state $s_t$. A ccording to $\varepsilon$ -greedy policy, this probability is given by [37]

$$\pi(r_i, s_t, a_{t,k}) = \begin{cases} 1 - \dfrac{\varepsilon}{|A_t|} & if \ \ Q(r_i, s_t, a_{t,k}) = \underset{\forall a_{t,j} \in A_t}{max} Q(r_i, s_t, a_{t,j}) \\[3mm] \dfrac{\varepsilon}{|A_t|} & otherwise \end{cases} \tag{12}$$

where $|A_t|$ denotes the size of the set $A_t$, that is, the number of actions that the mesh router $r_i$ can select.

**Algorithm 1** The pseudo-code of the reinforcement learning algorithm for solving RNP problem

**Input:**

- Network area ($W \times H$);
- The set of mesh clients ($C = \{c_i | i = 1..n\}$), and the set of its coordinates ($P_c = \{(x_{c_i}, y_{c_i}) | i = 1..n\}$);
- The set of gateway routers ($W = \{w_i | i = 1..k\}$), and the set of its coordinates ($P_w = \{(x_{w_i}, y_{w_i}) | i = 1..k\}$);
- The set of mesh routers ($R = \{r_i | i = 1..m\}$), and the coverage radius of each mesh router ($d_r$);

**Output:** The set of the best coordinates of $m$ mesh routers: $P_r = \{(x_{r_i}, y_{r_i}) | i = 1..m\}$

**Method:**
1: Place $m$ mesh routers at the coordinates $(x_{r_i}, y_{r_i}), i = 1..m$, where $x_{r_i}$ and $y_{r_i}$ are random values in the area $W \times H$;
2: **while** ($learn \le numLearn$) **do**
3: Randomly choose mesh router $r_i \in R$;
4: **for** (*each action $a_j \in A$*) **do**
5: Update $Q(r_i, s_t, a_{t,j})$ using (11);
6: **end for**
7: Choose the action $a_{t,k} \in A_t$ using policy derived from $Q$-values (e.g., $\varepsilon$ -greedy) according to (12);
8: Take action $a_{t,k}$, observe reward $R(r_i, s_t, a_{t,k})$ and next state $s_{t+1}$;
9: Update next state ($s_{t+1}$) to current state ($s_t$);
10: $learn \leftarrow learn + 1$;
11: **end while**
12: $P_r \leftarrow P_r$ in state $s_t$;

**Analyze the computational complexity.** The computational complexity of Algorithm 1 depends mainly on the iteration in Step (2), the number of possible actions in Step (4), and the algorithm for updating the Q value in Step (5). $Q(r_i, s_t, a_{t,j})$ is updated using Eq. (11), where the greatest complexity is the calculation of $RW(r_i, s_t, a_t)$ according to (10). $RW(r_i, s_t, a_t)$ contains two metrics, *CRR* and *CCR*, which are defined by (1) and (5), respectively. To determine *CRR*, we employed a breadth-first search algorithm on a network of $m$ vertices, which is the number of mesh routers. Therefore, the computational complexity was $O(m^2)$. The *CCR* is calculated using two nested loops of sizes $m$ and $n$, where $n$ is the number of mesh clients. Therefore, the complexity was $O(m \times n)$. Because $n$ is always greater than $m$ in a WMN, the computational complexity of $RW(r_i, s_t, a_t)$ is $O(m \times n)$. Consequently, the computational complexity of Algorithm 1 is $O(I \times |A| \times m \times n)$, where $I$ is the number of iterations and $|A|$ is the number of possible actions.

The computational complexity of Algorithm 1 is greater than that of the algorithms solving the RNP problem using GA [40], PSO [24], and WOA [41], which we compare in the following section. However, because its computing complexity is a polynomial function, it can be implemented in practice. Furthermore, because the algorithms for solving the RNP problem are run offline, the polynomial complexity is acceptable.

## Simulation results and discussion

### Simulation scenarios

The performance of the proposed method was evaluated through a simulation using Python. Our proposed method is compared with the most recent methods that use approximate optimization algorithms to address the RNP problem, including GA [40], PSO [24], WOA [41], and MVO [11]. All experiments were run on a 3.6 GHz Core i7 CPU computer. The surveyed network instances (NI) are presented in Table 3. NI-1 and NI-2 were used to investigate the effect of the number of mesh routers on the network performance, with the number of mesh routers ranging from 20 to 45 covering 150 mesh clients (NI-1) and 350 mesh clients (NI-2). NI-3 and NI-4 ware used to study the effect of client density, varying from 100 to 400. In NI-5 and N-6, the effect of the coverage radius of each mesh router was thoroughly examined. The final two NIs were used to investigate the influence of the network area. The parameters of the simulation scenarios and algorithms are presented in Table 4, where th parameters of the GA, PSO, WOA, and MVO are set as in [11].

### Simulation results

**Topology evaluation.** First, we evaluate the topology obtained when solving the RNP problem using the GA, PSO, WOA, MVO, and our proposed method, which employs

**Table 3. Network instances use for validating our proposed method.**

| No. | Routers | Clients | CR (m) | Network area | Purpose |
|---|---|---|---|---|---|
| NI-1 | 20:5:45 | 150 | 200 | 2000 × 2000 | Study the effect of the number of the mesh routers |
| NI-2 | 20:5:45 | 350 | 200 | 2000 × 2000 | |
| NI-3 | 30 | 100:50:400 | 200 | 2000 × 2000 | Study the effect of the number of the mesh clients |
| NI-4 | 45 | 100:50:400 | 200 | 2000 × 2000 | |
| NI-5 | 30 | 150 | 150:50:300 | 2000 × 2000 | Study the effect of the number of the coverage radius of the mesh routers |
| NI-6 | 45 | 350 | 150:50:300 | 2000 × 2000 | |
| NI-7 | 40 | 250 | 200 | 2000:500:3000 × 2000:500:3000 | Study the effect of the number of the network area |
| NI-8 | 45 | 250 | 200 | 2000:500:3000 × 2000:500:3000 | |

**Table 4. The parameters of algorithms and scenarios.**

| Parameter | Setting | Use for |
|---|---|---|
| Learning rate factor ($\alpha$) | 0.7 | Q-Learning algorithm |
| Discount factor ($\gamma$) | 0.5 | |
| $\varepsilon$ factor of greedy policy | 0.1 | |
| Step of actions | 10 [m] | |
| Population size | 50 | GA algorithm |
| Crossover Rate | 0.7 | |
| Mutation Rate | 0.01 | |
| Population size | 50 | PSO algorithm |
| $c_1$ | 2 | |
| $c_2$ | 2 | |
| Inertia weight | 1 | |
| Search-agent Number | 50 | WOA algorithm |
| $a$ | Decrease from 2 to 0 | |
| Universes | 50 | MVO algorithm |
| WEP | Increase from 0.2 to 1 | |
| TDR | Decrease from 0.6 to 0 | |
| $\lambda$ | 0.5 | Simulation scenarios |
| Number of runs | 50 | |
| Number of iteration | 2000 | |

reinforcement learning. The results obtained in Fig 3 clearly show topological differences between the methods. These findings were obtained using NI-2 with 30 mesh routers covering 350 mesh clients in an area of $2000 \times 2000$ [$m^2$] and a coverage radius of 200 [$m$] for each mesh router. We can observe that the method using RL provides the most optimal topology compared with the methods using approximate optimization algorithms, GA, WOA, PSO, and MVO. Specifically, for the method using reinforcement learning, there are 334 mesh clients covered by at least one mesh router, corresponding to a rate of 95.43%. These values were 292 (83.43%), 309 (88.29%), 313 (89.43%), and 313 (88.86%) for the WOA, GA, PSO, and MVO algorithms, respectively. In addition, the topology of the reinforcement learning method has a wider coverage area than the other methods, which can increase the percentage of clients covered in the case of denser clients.

**Impact of mesh router density.** In this section, the impact of the mesh router density on network performance is investigated using various simulation scenarios. We use the most important metric often used to evaluate the performance of RNP problem solving methods, that is, network connectivity (NC). In our context, the NC is calculated as

$$NC \ (\%) = \frac{\sum_{i=1}^{m} \alpha(r_i) + \sum_{j=1}^{n} \beta(c_j)}{m + n} \times 100 \tag{13}$$

where $\alpha(r_i)$ and $\beta(c_j)$ are determined according to (2) and (3), respectively, $m$ and $n$ represent the number of mesh routers and mesh clients, respectively.

The results obtained in Fig 4 clearly show the difference in network connectivity between the proposed method and the method using approximate optimization algorithms. These findings were obtained using NI-1, in which the number of mesh routers varieed from 20 to 45, covering 150 mesh clients in an area of $2000 \times 2000$ [$m^2$] and a coverage radius of 200 [$m$] for each mesh router. We can observe that the NC increases proportionally with the number of mesh routers for all methods. This is evident because as the number of mesh routers increases,

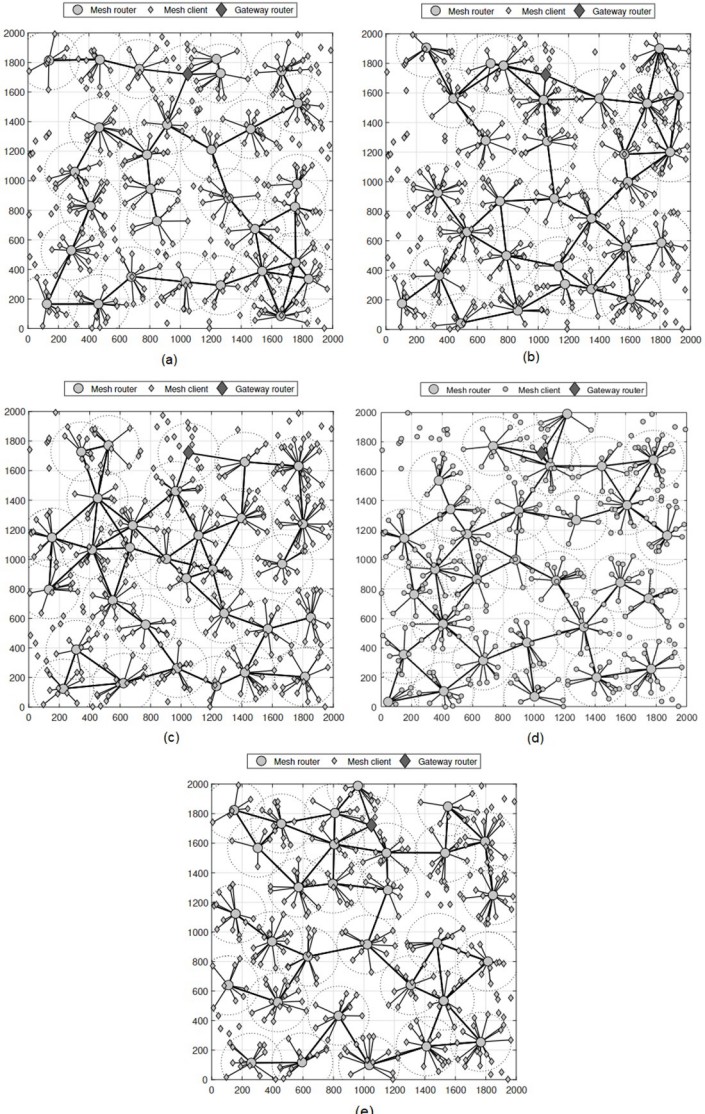

**Fig 3. Compare WMN topologies using different router node placement algorithms.** (a) WOA, (b) GA, (c) PSO, (d) MVO, and (e) reinforcement learning.

the coverage area expands, increasing the probability of mesh clients being covered. Comparing the methods of solving RNP problems, the method using RL (legend namely RL-based RNP) gives the highest NC. For example, considering the case of 35 mesh routers, The NC values of the methods using the WOA, PSO, GA, MVO, and RL are 85.64, 87.42, 90.67, 93.42, and 95.68%, respectively. Thus, compared with the method using algorithms WOA, PSO, GA, and MVO, the proposed method improved NC by 10.03, 8.25, 5.01%, and 2.25%, respectively. This is a significant result in improving WMN performance.

The results obtained were quite similar for the implementation on NI-2, as shown in Fig 5. The assumptions of this simulation scenario are the same as those in NI-1, except that the number of mesh clients increases to 350. We can see that the proposed method is highly effective in terms of NC. We can observe that the proposed solution provides high efficiency in terms of NC for most values of the number of mesh routers. The NC of the method using RL

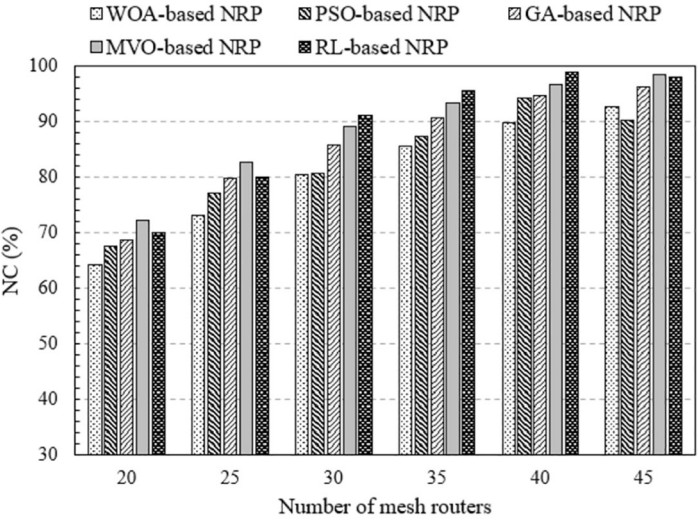

**Fig 4. Evaluate the network connectivity versus the number of mesh routers using NI-1.**

increases by an average of 4 to 20% compared with the cases where approximate optimization algorithms are used. As is the case with 35 mesh routers, the NC of the RL is 98.71%. These values of the WOA, PSO, GA, and MVO algorithms were 81.66%, 86.89%, 88.59%, and 94.44% respectively. Thus, the method using RL improved the NC from 4.26% to 17.04%.

Based on the findings in Figs 4 and 5, we can conclude that changing the number of mesh routers affects on network performance in terms of NC. The larger the number of routers, the higher the NC for all investigated RNP problem solving methods. In particular, the method based on RL is the most efficient.

**Impact of mesh client density.**  In this section, we investigate the effect of client density on network performance. In a WMN, the denser the clients, the greater is the number of connection requests to the routers. As a result, network performance was affected. This is more

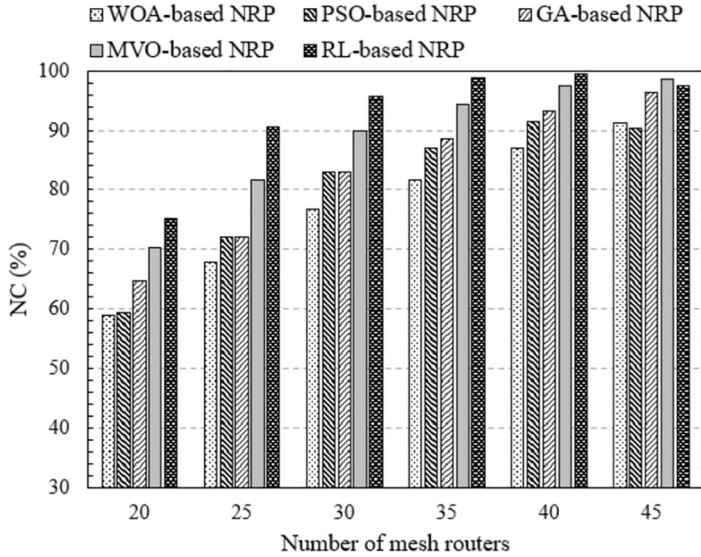

**Fig 5. Evaluate the network connectivity versus the number of mesh routers using NI-2.**

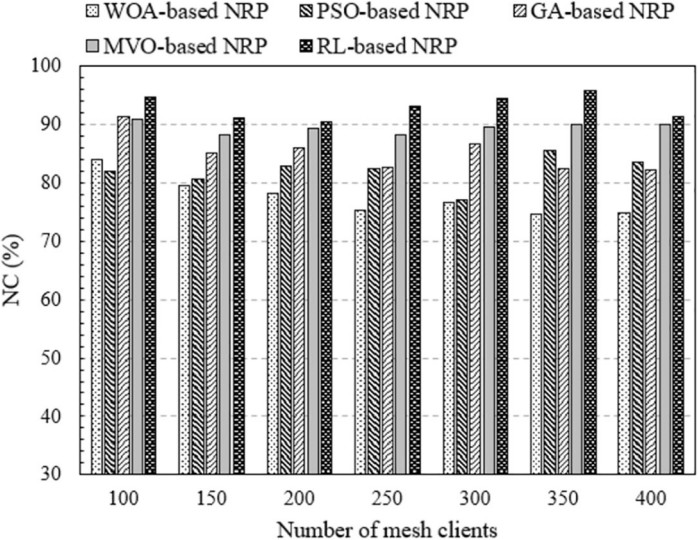

**Fig 6. Evaluate the network connectivity versus the number of mesh clients using NI-3.**

evident in Fig 6, where we plot NC as a function of the number of mesh clients. These results are obtained by executing NI-3, where the number of mesh routers is 30, covering 150 to 300 mesh clients. We can easily observe that the method using RL always yields the highest NC regardless of whether the client density is sparse or dense. The NC value of this method from 90.43% to 95.79%. Meanwhile, the NC values for the cases of algorithm WOA, PSO, GA, and MVO are fom 74.59% to 84.08%, from 77.00% to 85.63%, from 82.32% to 91.46%, and from 88.27% to 90.83%, respectively. When 45 mesh routers were used (NI-4), the NC value increased for all methods. This is clearly shown in Fig 7, where we represent NC versus the number of mesh clients. Comparing the methods, we find that the method using RL outperforms the method using approximate optimal algorithms in terms of NC.

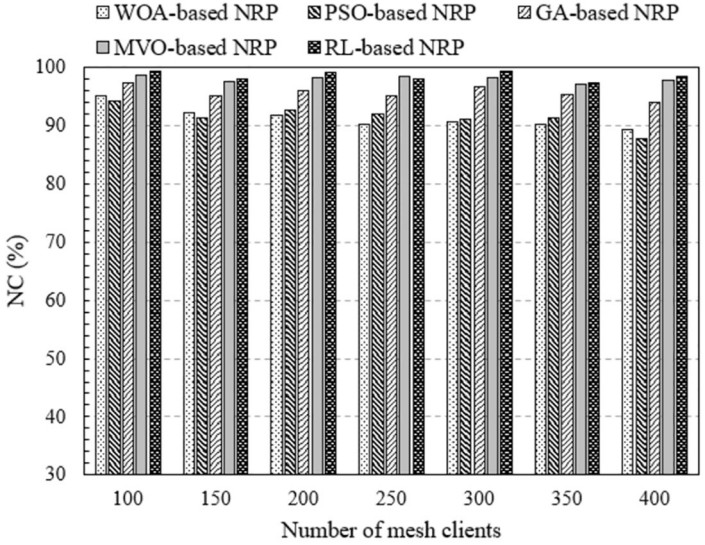

**Fig 7. Evaluate the network connectivity versus the number of mesh clients using NI-4.**

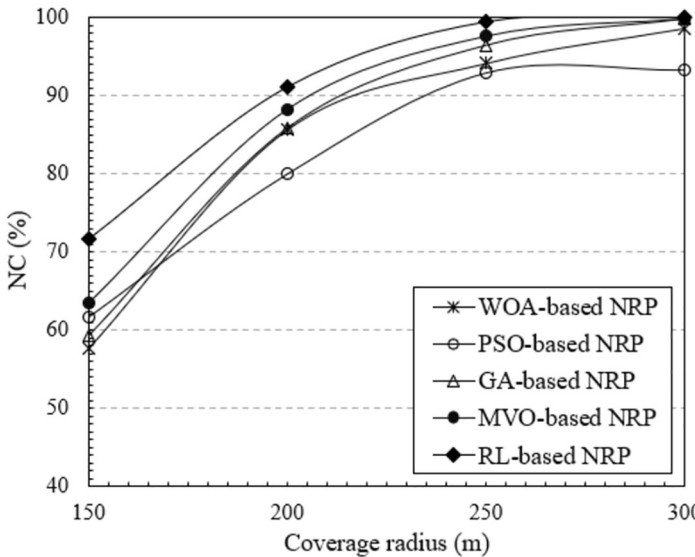

**Fig 8. Evaluate the network connectivity versus the coverage radius of mesh routers using NI-5.**

**Impart of the coverage radius of mesh routers.** The coverage radius of the mesh routers is another technological parameter that has a considerable impact on the WMN performance. In this section, we investigate the effect of this technological parameter on the NC metric. The results obtained in Fig 8 clearly show the change in NC with respect to the coverage radius of the mesh routers. These results were implemented using NI-5, which has 30 mesh routers and 150 mesh clients. The coverage radius of each router ranged from 150 to 300 [$m$]. The plots in Fig 8 indicate that the NC increases proportionally to the coverage radius of the mesh routers. This is because expanding the coverage radius increases the likelihood that clients will be covered. As a result, NC increases. In particular, the method using RL yielded the highest NC, reaching close to 100% when the coverage radius was 250 [m] or more. The results are also similar for NI-6, as shown in Fig 9. The NC value of this NI is greater than that of the NI-5

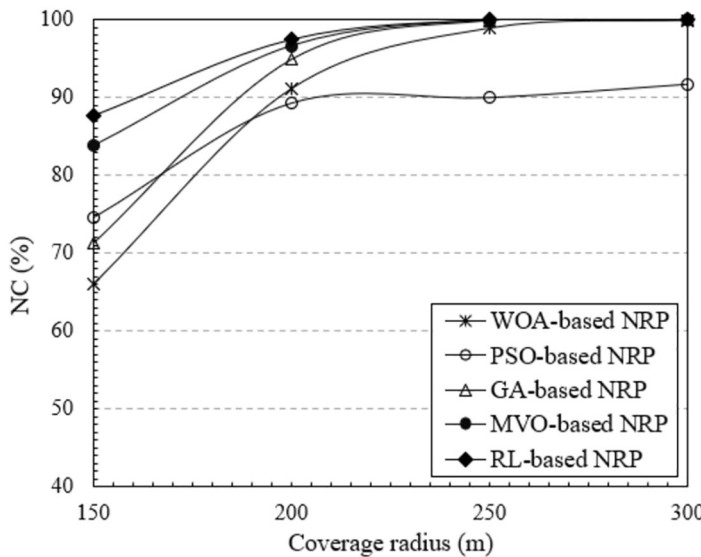

**Fig 9. Evaluate the network connectivity versus the coverage radius of mesh routers using NI-6.**

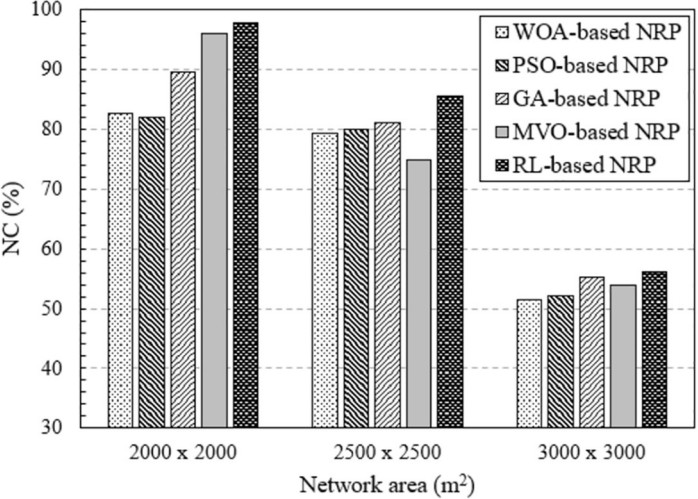

**Fig 10. Evaluate the network connectivity versus network area using NI-7.**

because this uses more mesh routers. As in the previous scenarios, the method using RL always yields the highest NC.

   **Impact of network area.**   In the last section, we investigate the effect of network area on the efficiency of RNP problem solving methods. Figs 10 and 11 show the results obtained by executing NI-7 and NI-8, respectively. In these NIs, the network area varies from $2000 \times 2000$ [$m^2$] to $3000 \times 3000$ [$m^2$]. The NC value decreased according to the network area for all the algorithms. This is because, for a given number of mesh routers, the larger the network area, the lower the percentage of area covered, leading to a decrease in the NC value. However, the NC value of the method using RL is always the largest.

   Based on the above findings, we can conclude that the proposed method, which uses reinforcement learning to solve the RNP problem, is more efficient than a method that uses approximate optimal algorithms. This is a crucial result in the design and implementation of a

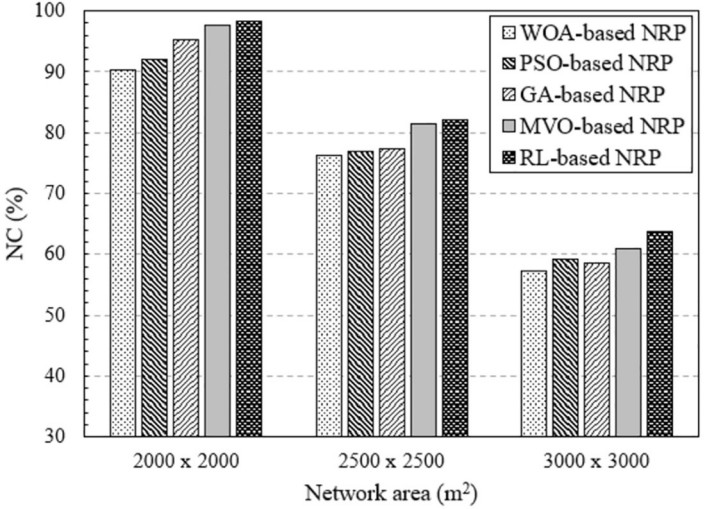

**Fig 11. Evaluate the network connectivity versus network area using NI-8.**

WMN, which helps find an optimal network topology to exploit network resources more efficiently.

## Conclusion

The placement of router nodes in wireless mesh networks is a significant problem that has recently attracted the interest of several research groups. This problem is recognized as NP-hard, and cannot be resolved using conventional algorithms. In this study, we proposed a new and effective method for solving this problem using RL. The process of finding the optimal coordinates for placing mesh routers is modeled as an RL with the main components being environment, agent, action, and reward, which are equivalent to the network system, routers, coordinate adjustment, and network connectivity of the RNP problem, respectively. Simulation results show that our proposed method outperforms the most recent methods in terms of coverage and network connectivity.

In future work, we will continue to develop this method by considering additional constraints on the quality of transmission and load balancing to improve network performance. In addition, the deep reinforcement learning method can also be applied to static and dynamic RNP problems to further improve the performance of the WMN.

## Supporting information

**S1 Dataset.**
(ZIP)

## Author Contributions

**Conceptualization:** Thuy-Van T. Duong.

**Investigation:** Le Huu Binh, Thuy-Van T. Duong.

**Methodology:** Le Huu Binh, Thuy-Van T. Duong.

**Software:** Le Huu Binh.

**Writing – original draft:** Le Huu Binh.

**Writing – review & editing:** Le Huu Binh, Thuy-Van T. Duong.

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
