## [Decision Letter · Decision Letter 0]

20 Sep 2023

PONE-D-23-25280A Novel and Effective Method for solving the Router Nodes Placement in Wireless Mesh Networks using Reinforcement LearningPLOS ONE

Dear Dr. T. Duong,

Thank you for submitting your manuscript to PLOS ONE. After careful consideration, we feel that it has merit but does not fully meet PLOS ONE’s publication criteria as it currently stands. Therefore, we invite you to submit a revised version of the manuscript that addresses the points raised during the review process.

We look forward to receiving your revised manuscript.

Kind regards,

Mohammed Balfaqih

Academic Editor

PLOS ONE

ournal requirements:

4. PLOS requires an ORCID iD for the corresponding author in Editorial Manager on papers submitted after December 6th, 2016. Please ensure that you have an ORCID iD and that it is validated in Editorial Manager. To do this, go to ‘Update my Information’ (in the upper left-hand corner of the main menu), and click on the Fetch/Validate link next to the ORCID field. This will take you to the ORCID site and allow you to create a new iD or authenticate a pre-existing iD in Editorial Manager. Please see the following video for instructions on linking an ORCID iD to your Editorial Manager account: " ext-link-type="uri" xlink:type="simple">https://www.youtube.com/watch?v=_xcclfuvtxQ".

5. We note that Figure 1, 2, and 3 in your submission contain copyrighted images. All PLOS content is published under the Creative Commons Attribution License (CC BY 4.0), which means that the manuscript, images, and Supporting Information files will be freely available online, and any third party is permitted to access, download, copy, distribute, and use these materials in any way, even commercially, with proper attribution. For more information, see our copyright guidelines: http://journals.plos.org/plosone/s/licenses-and-copyright.

A. You may seek permission from the original copyright holder of Figure 1, 2, and 3 to publish the content specifically under the CC BY 4.0 license. 

B. If you are unable to obtain permission from the original copyright holder to publish these figures under the CC BY 4.0 license or if the copyright holder’s requirements are incompatible with the CC BY 4.0 license, please either i) remove the figure or ii) supply a replacement figure that complies with the CC BY 4.0 license. Please check copyright information on all replacement figures and update the figure caption with source information. If applicable, please specify in the figure caption text when a figure is similar but not identical to the original image and is therefore for illustrative purposes only.

7. We are unable to open your Supporting Information file [LateXSourceFile.zip]. Please kindly revise as necessary and re-upload.

Reviewers' comments:

Reviewer's Responses to Questions

**Comments to the Author**

1. Is the manuscript technically sound, and do the data support the conclusions?

Reviewer #1: Yes

Reviewer #2: Yes

2. Has the statistical analysis been performed appropriately and rigorously? 

Reviewer #1: Yes

Reviewer #2: Yes

3. Have the authors made all data underlying the findings in their manuscript fully available?

Reviewer #1: Yes

Reviewer #2: Yes

4. Is the manuscript presented in an intelligible fashion and written in standard English?

Reviewer #1: No

Reviewer #2: Yes

5. Review Comments to the Author

Reviewer #1: After a careful reading of the manuscript, some of my concerns are listed below:

1. Usage of English should be improved. Especially in the abstract and Introduction section, the authors should provide more details on the motivation behind the work.

2. Figures should be included in the main text. Further, the quality of figure is not good. High defination images should be used.

3. The author should justify in the introduction section that how the proposed work is unique and superior than the following literature:

[a] Ren, Haoxing, and Matthew Fojtik. "Nvcell: Standard cell layout in advanced technology nodes with reinforcement learning." In 2021 58th ACM/IEEE Design Automation Conference (DAC), pp. 1291-1294. IEEE, 2021.

[b] Sharma, Anamika, and Siddhartha Chauhan. "A distributed reinforcement learning based sensor node scheduling algorithm for coverage and connectivity maintenance in wireless sensor network." Wireless Networks 26, no. 6 (2020): 4411-4429.

[c] Ouamri, Mohamed Amine, Gordana Barb, Daljeet Singh, and Florin Alexa. "Load balancing optimization in software-defined wide area networking (SD-WAN) using deep reinforcement learning." In 2022 International Symposium on Electronics and Telecommunications (ISETC), pp. 1-6. IEEE, 2022.

[d] Taleb, Sylia Mekhmoukh, Yassine Meraihi, Seyedali Mirjalili, Dalila Acheli, Amar Ramdane-Cherif, and Asma Benmessaoud Gabis. "Mesh Router Nodes Placement for Wireless Mesh Networks Based on an Enhanced Moth–Flame Optimization Algorithm." Mobile Networks and Applications (2023): 1-24.

[e] Ouamri, Mohamed Amine, Mohamed Azni, Daljeet Singh, Waleed Almughalles, and Mohammed Saleh Ali Muthanna. "Request delay and survivability optimization for software defined‐wide area networking (SD‐WAN) using multi‐agent deep reinforcement learning." Transactions on Emerging Telecommunications Technologies (2023): e4776.

4. Required citation to the literature should be given for mathematical expressions in the paper. For example, for Connected router ratio and Connected client ratio.

5. A comparison of computational complexity should be presented to fully justify the proposed algorithm.

Reviewer #2: The paper discusses the challenge of Router Node Placement (RNP) in wireless mesh networks, a known NP-hard problem. Traditional algorithms cannot efficiently solve this problem, so the paper introduces a novel approach using reinforcement learning. They model RNP as a reinforcement learning problem with network components representing the environment, routers, coordinate adjustment, and connectivity. This study is the first to apply reinforcement learning to RNP. Experimental results demonstrate that their method surpasses recent methods in terms of network coverage and connectivity.

1. Although the authors have discussed some essential background, I still do not well catch the main novelty directly. The challenges, methodology, and improvements need to be better clarified, both from research and application aspects, in the Introduction section.

2. The novelty of this paper needs to be highlighted.

3. The motivation to the problem need to be discussed in the abstract. Why authors are considering RL instead of Approximation or Heuristics.

4. Figures should keep in suitable location.

5. There are typos, grammatical errors, punctuation errors and other mistakes. Some sentences are not clear.

For example:

"In addition, there is at least one the router of the internet"

6. What protocol is used to communicate when two routers are within the range of each other.

7. Authors have to clarify whether they are comparing the proposed algorithm with the approximation algorithms or Heuristic algorithms.

8. GA, PSO and WoA are not approximation algorithms.

9. The authors have to consider the dynamic algorithms instead of static algorithms

10. The proposed algorithm is RL type, it is suggested that the proposed algorithm need to be compared with the RL algorithms only instead of comparing it with heuristic algorithms.

11. Perform the comparative analysis with the latest works based on RL related placement problems.

A) "Deep reinforcement learning mechanism for deadline-aware cache placement in device-to-device mobile edge networks." Wireless Networks (2022): 1-20.

B) "Mesh Router Nodes Placement for Wireless Mesh Networks Based on an Enhanced Moth–Flame Optimization Algorithm." Mobile Networks and Applications, 1-24.

C) "An efficient mesh router nodes placement in wireless mesh networks based on moth‐flame optimization algorithm". International Journal of Communication Systems, 36(8), e5468.

12. I suggest the authors include a subsection to show the complexity (Complexity of the proposed scheme), convergence, and optimality.

6. PLOS authors have the option to publish the peer review history of their article (what does this mean?). If published, this will include your full peer review and any attached files.

Reviewer #1: **Yes: **Daljeet Singh

Reviewer #2: No

---

## [Author Response · Author response to Decision Letter 0]

7 Dec 2023

Dear Editors and Reviewers,

On behalf of the authors, I would like to thank you for reviewing our manuscript entitled “A Novel and Effective Method for solving the Router Nodes Placement in Wireless Mesh Networks using ReinforcementLearning”. The comments of the Editors and Reviewers are valuable and exceptionally supportive of revising and improving the manuscript. We have carefully considered these comments and revised them accordingly.

We look forward to hearing from the editor and the reviewers. Please address all correspondence concerning this manuscript to me at duongthithuyvan@tdtu.edu.vn. 

Thank you for your consideration of this manuscript.

Sincerely,

Thuy-Van T. Duong et. al.

---

## [Decision Letter · Decision Letter 1]

18 Jan 2024

PONE-D-23-25280R1A Novel and Effective Method for solving the Router Nodes Placement in Wireless Mesh Networks using Reinforcement LearningPLOS ONE

Dear Dr. T. Duong,

Thank you for submitting your manuscript to PLOS ONE. After careful consideration, we feel that it has merit but does not fully meet PLOS ONE’s publication criteria as it currently stands. Therefore, we invite you to submit a revised version of the manuscript that addresses the points raised during the review process. Please submit your revised manuscript by Mar 04 2024 11:59PM. If you will need more time than this to complete your revisions, please reply to this message or contact the journal office at plosone@plos.org. Please include the following items when submitting your revised manuscript:A rebuttal letter that responds to each point raised by the academic editor and reviewer(s). You should upload this letter as a separate file labeled 'Response to Reviewers'.A marked-up copy of your manuscript that highlights changes made to the original version. You should upload this as a separate file labeled 'Revised Manuscript with Track Changes'.An unmarked version of your revised paper without tracked changes. You should upload this as a separate file labeled 'Manuscript'.If applicable, we recommend that you deposit your laboratory protocols in protocols.io to enhance the reproducibility of your results. Protocols.io assigns your protocol its own identifier (DOI) so that it can be cited independently in the future. For instructions see: https://journals.plos.org/plosone/s/submission-guidelines#loc-laboratory-protocols. Additionally, PLOS ONE offers an option for publishing peer-reviewed Lab Protocol articles, which describe protocols hosted on protocols.io. Read more information on sharing protocols at https://plos.org/protocols?utm_medium=editorial-emailutm_source=authorlettersutm_campaign=protocols.

We look forward to receiving your revised manuscript.

Kind regards,

Mohammed Balfaqih

Academic Editor

PLOS ONE

Journal Requirements:

Reviewers' comments:

Reviewer's Responses to Questions

**Comments to the Author**

1. If the authors have adequately addressed your comments raised in a previous round of review and you feel that this manuscript is now acceptable for publication, you may indicate that here to bypass the “Comments to the Author” section, enter your conflict of interest statement in the “Confidential to Editor” section, and submit your "Accept" recommendation.

Reviewer #2: All comments have been addressed

Reviewer #3: All comments have been addressed

2. Is the manuscript technically sound, and do the data support the conclusions?

Reviewer #2: Yes

Reviewer #3: No

3. Has the statistical analysis been performed appropriately and rigorously? 

Reviewer #2: Yes

Reviewer #3: I Don't Know

4. Have the authors made all data underlying the findings in their manuscript fully available?

Reviewer #2: No

Reviewer #3: No

5. Is the manuscript presented in an intelligible fashion and written in standard English?

Reviewer #2: Yes

Reviewer #3: No

6. Review Comments to the Author

Reviewer #2: All of my concerns have been addressed. I congratulate the authors and All the best.

I recommend this work for possible publication. (Accepted)

The summer of the manuscript is:

The paper discusses the challenge of Router Node Placement (RNP) in wireless mesh networks, a known NP-hard problem. Traditional algorithms cannot efficiently solve this problem, so the paper introduces a novel approach using reinforcement learning. They model RNP as a reinforcement learning problem with network components representing the environment, routers, coordinate adjustment, and connectivity. This study is the first to apply reinforcement learning to RNP. Experimental results demonstrate that their method surpasses recent methods in terms of network coverage and connectivity.

Reviewer #3: This paper presents a novel and effective method for solving the router nodes placement in wireless mesh networks using reinforcement learning.

1- The writing of paper should be improved seriously, equations should be cited and the content should be justify

2- Please add the improvement of your method results in the abstract numerically.

3- Please compare your paper with the state-of-the art papers:

a. RLS2: An energy efficient reinforcement learning-based sleep scheduling for energy harvesting WBANs, by R Mohammadi…

b. DRDC: Deep reinforcement learning based duty cycle for energy harvesting body sensor node, by by R Mohammadi…

4- you should use more approximation algorithms.

7. PLOS authors have the option to publish the peer review history of their article (what does this mean?). If published, this will include your full peer review and any attached files.

Reviewer #2: No

Reviewer #3: No

---

## [Author Response · Author response to Decision Letter 1]

18 Feb 2024

Reviewer #2: All of my concerns have been addressed. I congratulate the authors and All the best.

I recommend this work for possible publication. (Accepted)

The summer of the manuscript is:

The paper discusses the challenge of Router Node Placement (RNP) in wireless mesh networks, a known NP-hard problem. Traditional algorithms cannot efficiently solve this problem, so the paper introduces a novel approach using reinforcement learning. They model RNP as a reinforcement learning problem with network components representing the environment, routers, coordinate adjustment, and connectivity. This study is the first to apply reinforcement learning to RNP. Experimental results demonstrate that their method surpasses recent methods in terms of network coverage and connectivity.

Author response: Thank you very much for accepting the manuscript.

Reviewer #3: This paper presents a novel and effective method for solving the router nodes placement in wireless mesh networks using reinforcement learning.

1- The writing of paper should be improved seriously, equations should be cited and the content should be justify

Author response: Thank you for your valuable comments. We have reviewed and edited the manuscript, all equations have been fully cited except those proposed by us. The content is aligned to the left because it follows the template of PLOS ONE.

2- Please add the improvement of your method results in the abstract numerically.

Author response: Thank you for your valuable comments. We have added to the abstract the numerical value of increased network connectivity of the proposed method compared to the most recent methods.

3- Please compare your paper with the state-of-the art papers:

a. RLS2: An energy efficient reinforcement learning-based sleep scheduling for energy harvesting WBANs, by R Mohammadi…

b. DRDC: Deep reinforcement learning based duty cycle for energy harvesting body sensor node, by by R Mohammadi…

Thank you for introducing new articles of high scientific and practical significance. We investigated these works and have cited related work (ref. [38, 39]) on the topic of applications of reinforcement learning in wireless networks (page 6).

Since the problem in our work is different from the problem in those works (RLS2 and DRDC), we cannot compare our work with them. We can develop suitable research topics to compare with these works in the future.

4- you should use more approximation algorithms.

Author response: Thank you for your valuable comments. We used an additional approximation algorithm, which is the Multi-Verse Optimizer (MVO) algorithm to compare with the proposed algorithm. The experimental results have been updated in Figures 3 to 11 and the text analyzing the results of these figures.

---

## [Decision Letter · Decision Letter 2]

11 Mar 2024

A Novel and Effective Method for solving the Router Nodes Placement in Wireless Mesh Networks using Reinforcement Learning

PONE-D-23-25280R2

Dear Dr. T. Duong,

We’re pleased to inform you that your manuscript has been judged scientifically suitable for publication and will be formally accepted for publication once it meets all outstanding technical requirements.

Kind regards,

Mohammed Balfaqih

Academic Editor

PLOS ONE

Additional Editor Comments (optional):

Reviewers' comments:

Reviewer's Responses to Questions

**Comments to the Author**

1. If the authors have adequately addressed your comments raised in a previous round of review and you feel that this manuscript is now acceptable for publication, you may indicate that here to bypass the “Comments to the Author” section, enter your conflict of interest statement in the “Confidential to Editor” section, and submit your "Accept" recommendation.

Reviewer #3: All comments have been addressed

2. Is the manuscript technically sound, and do the data support the conclusions?

Reviewer #3: Yes

3. Has the statistical analysis been performed appropriately and rigorously? 

Reviewer #3: Yes

4. Have the authors made all data underlying the findings in their manuscript fully available?

Reviewer #3: Yes

5. Is the manuscript presented in an intelligible fashion and written in standard English?

Reviewer #3: Yes

6. Review Comments to the Author

Reviewer #3: Dear Editor

All of comments are applied, Thanks, Please check grammar ones again ...............................................

7. PLOS authors have the option to publish the peer review history of their article (what does this mean?). If published, this will include your full peer review and any attached files.

Reviewer #3: No

---

## [Editor Report · Acceptance letter]

28 Mar 2024

PONE-D-23-25280R2 

PLOS ONE

Dear Dr. T. Duong, 

I'm pleased to inform you that your manuscript has been deemed suitable for publication in PLOS ONE. Congratulations! Your manuscript is now being handed over to our production team.

Kind regards, 

on behalf of

Dr. Mohammed Balfaqih 

Academic Editor

PLOS ONE